# Comprehensive Serum Glycopeptide Spectra Analysis (CSGSA): A Potential New Tool for Early Detection of Ovarian Cancer

**DOI:** 10.3390/cancers11050591

**Published:** 2019-04-27

**Authors:** Masaru Hayashi, Koji Matsuo, Kazuhiro Tanabe, Masae Ikeda, Mariko Miyazawa, Miwa Yasaka, Hiroko Machida, Masako Shida, Tadashi Imanishi, Brendan H. Grubbs, Takeshi Hirasawa, Mikio Mikami

**Affiliations:** 1Department of Obstetrics and Gynecology, Tokai University School of Medicine, Kanagawa 2591193, Japan; hayashimasaru0620@hotmail.com (M.H.); ikedam@tokai-u.jp (M.I.); marikom@tokai-u.jp (M.M.); mm_4amm1302@yahoo.co.jp (M.Y.); hiroko.machida@tokai.ac.jp (H.M.); shida@is.icc.u-tokai.ac.jp (M.S.); hira@is.icc.u-tokai.ac.jp (T.H.); 2Division of Gynecologic Oncology, Department of Obstetrics and Gynecology, University of Southern California, Los Angeles, CA 90033, USA; koji.matsuo@med.usc.edu; 3Norris Comprehensive Cancer Center, University of Southern California, Los Angeles, CA 90033, USA; 4Medical Solution Promotion Department, LSI Medience Corporation, Tokyo 1748555, Japan; tanabe.kazuhiro@mp.medience.co.jp; 5Department of Molecular Life Science, Division of Basic Medical Science and Molecular Medicine, Tokai University School of Medicine, Kanagawa 2591193, Japan; imanishi@tokai-u.jp; 6Division of Maternal-Fetal Medicine, Department of Obstetrics and Gynecology, University of Southern California, Los Angeles, CA 90027, USA; brendan.grubbs@med.usc.edu

**Keywords:** comprehensive serum glycopeptide spectra analysis, orthogonal partial square discrimination analysis, epithelial ovarian cancer, ovarian clear cell carcinoma, endometrioma

## Abstract

Objectives: To conduct a comprehensive glycopeptide spectra analysis of serum between cancer and non-cancer patients to identify early biomarkers of epithelial ovarian cancer (EOC). Methods: Approximately 30,000 glycopeptide peaks were detected from the digested serum glycoproteins of 39 EOC patients (23 early-stage, 16 advanced-stage) and 45 non-cancer patients (27 leiomyoma and ovarian cyst cases, 18 endometrioma cases) by liquid chromatography mass spectrometry (LC–MS). The differential glycopeptide peak spectra were analyzed to distinguish between cancer and non-cancer groups by employing multivariate analysis including principal component analysis (PCA), orthogonal partial least squares discriminant analysis (OPLS-DA) and heat maps. Results: Examined spectral peaks were filtered down to 2281 serum quantitative glycopeptide signatures for differentiation between ovarian cancer and controls using multivariate analysis. The OPLS-DA model using cross-validation parameters R2 and Q2 and score plots of the serum samples significantly differentiated the EOC group from the non-cancer control group. In addition, women with early-stage clear cell carcinoma and endometriomas were clearly distinguished from each other by OPLS-DA as well as by PCA and heat maps. Conclusions: Our study demonstrates the potential of comprehensive serum glycoprotein analysis as a useful tool for ovarian cancer detection.

## 1. Introduction

Epithelial ovarian cancer (EOC) is the leading cause of gynecologic cancer mortality, and is difficult to detect at an early stage [1,2]. Over 70% of women with EOC are diagnosed with advanced stage disease with a 5-year relative survival of less than 50% [2]. Current screening methods lack sufficient accuracy [3,4], and thus, there is an urgent need to develop new strategies for detection of early stage EOC to improve patient survival.

We previously identified a C4-binding protein with fully-sialylated N-glycans (FS-C4BP = A2160) as a novel EOC marker [5]. The FS-C4BP particularly distinguished early-stage ovarian clear cell carcinoma (OCCC) from endometrioma more reliably than CA-125. In addition, A2160, as a FS-C4BP peptide with a fully sialylated and fucosylated triantennary glycan and fully sialylated biantennary glycan exhibited superior sensitivity and specificity as an EOC marker.

Common strategies for the discovery of new cancer biomarkers [6] include metabolomics [7,8], identification of genetic and genomic markers and gene mutations by PCR and/or next generation sequencing, anomalous alternative mRNA splicing products of an expressed gene, as well gene re-expression following silencing during normal differentiation [1,9]. The use of liquid chromatography with mass spectrometry (LC–MS) to identify glycoproteins released into the circulation provides a new approach to the discovery of cancer markers [5,10]. With this novel technology, we investigated not only sugar chain alterations but also combinations of protein and glycan changes [5].

The analysis of serum glycoproteins remains a difficult and exhausting procedure due to the large amount of data generated from the numerous peaks produced by the LC–MS method. High inter-individual variability and complexity in glycan and glycopeptide profiles makes the visual comparisons of these spectra impractical and multivariate analyses are required to determine consistent variations between data sets [11,12]. Some of the statistical methods required for the multivariate analysis of the instrumental spectral results involve bilinear and multiway models, cluster analysis, scatter plots, principal component analysis (PCA), linear and non-linear partial least squares, and pattern recognition computations through the newly developed application of orthogonal partial square discrimination analysis (OPLS-DA) in scatter and scores plots [13,14].

In our previous study, we identified the glycopeptide A2160 cancer marker from among more than 100,000 glycopeptide peaks detected in the blood sera of ovarian cancer patients [5]. During the course of detecting, isolating and identifying the A2160 glycopeptide we developed filtering procedures to find thousands of MS glycopeptide peaks that showed statistically significant differences between ovarian cancer patients and control patients. In this study, we investigated the comprehensive glycopeptide spectra analysis (CSGSA) of serum between cancer and non-cancer patients using the basic *t*-test, scatter plots, PCA, OPLS-DA, and heat maps in order to use the MS spectra as a possible novel approach for the early detection of ovarian cancer.

## 2. Materials and Methods

### 2.1. Patient Sample

Demographics are shown in Appendix A. During the study period from January to December of 2016, consecutive serum samples were obtained from 39 EOC patients prior to the initiation of any treatment. In addition, serum samples were collected from 45 subjects in a non-cancer group. The mean age of EOC patients was 53.7 years. The ethnicity of the EOC group was Japanese (100%) and the four most common histologic subtypes of EOC were OCCC (*n* = 15, 38.5%), endometrioid (*n* = 14, 35.8%), serous (*n* = 8, 20.5%), and mucinous adenocarcinoma (*n* = 2, 5.1%). The majority of cases were early-stage (stage I–II; 58.9%).

The non-cancer group included Japanese women with the following gynecologic diseases and mean ages (years): Uterine myoma (*n* = 10) in women with an average age of 51.0 years, endometrioma (*n* = 18) with an average age of 43.78 years, and ovarian cysts (*n* = 17) in women with an average age of 54.41 years.

All blood samples were collected by venipuncture from patients before their surgery. Samples were centrifuged and stored at −80 °C until needed, with avoidance of repeated freeze–thaw cycles. Samples were excluded from the study based on the patient criteria listed in Appendix A.

### 2.2. Quality Control Serum and Inter- and Intra-Assay Coefficients of Variability (CVs)

Quality control (QC) serum samples were prepared by separately pooling ten sera of ovarian cancer patients and ten sera of benign gynecologic disease patients. The pooled sera were preserved in −80 °C until they were analyzed. The QC samples were used during the process of glycoprotein profiling to calculate the inter- and intra-assay coefficients of variability (CVs) due to operator or laboratory generated variations and errors.

### 2.3. Sample Preparation for Glycoprotein Profiling

Mikami et al. [5] previously presented a description of the techniques and the schema that were used for the glycoprotein profiling in this study. Briefly, after extracting and protease digestion of the glycoproteins, the glycopeptides were enriched by filtration and analyzed by liquid chromatography mass spectrometry (LC–MS). The detailed sample preparation methods are described in Appendix A and are slightly modified from the previous report [5].

### 2.4. Liquid Chromatography and Mass Spectrometry

UPLC–MS/MS data were acquired on an UPLC system (Agilent HP1200; Agilent Technologies, Palo Alto, CA, USA) equipped with a C18 column (Inertsil ODS-4, 2 μm, 100 Å, 100 mm × 1.5 mm ID; GL Science, Tokyo, Japan) and coupled with an electrospray ionization quadrupole time-of-flight (Q-TOF) mass spectrometer (Agilent 6520, Agilent Technologies). Solvent A was 0.1% formic acid, and solvent B comprised 0.1% formic acid in 9.9% water and 90% acetonitrile. Glycopeptides were eluted at 40 °C with a flow rate of 0.15 mL/min, using the following gradient program: 0–7 min, 15–30% solvent B; 7–12 min, 30–50% solvent B; and an additional 2-min hold at 100% solvent B. The mass spectrometer was operated in negative mode with a capillary voltage of 4000 V. The nebulizing gas pressure was 30 psi, and the dry gas flow was 8 L/min at 350 °C. The injection volume was 5 μL.

### 2.5. Data Analysis

All mass spectral data were analyzed by using our original software, “Marker Analysis,” developed using R (R 3.2.2, R Foundation) and Excel VBA (Excel 2010, Microsoft, Washington, VA, USA) as previously reported [15]. After LC–MS raw data were converted to CSV-type data using Mass Hunter Export (Agilent Technologies), Marker Analysis was used to distinguish peak curve shapes, smooth and differentiate the peaks, and recognize the beginning, top, and end of them (determined at the points where the differentiation curve changed from zero to positive, from positive to negative, and from negative to zero, respectively). The peak area was calculated by integrating curves from beginning to end. The error in retention time and m/z was corrected using internal standard (fetal calf fetuin) peaks. Peak alignment was performed in such a way that the error of each peak position (retention time and m/z) was within 0.3 min and 0.06 Da, respectively.

The data was normalized by calculating ratios between each peak area and the average peak areas of QCs. The mode establishing method with SIMCA software (version 13.0.3; Umetrics; Sartorius AG, Göttingen, Germany) was used to form the orthogonal partial least-squares discriminant analysis (OPLS-DA) model [16]. Heat maps were prepared using the protocols developed for the software program Excel VBA.

### 2.6. Pattern Recognition (PR) Analysis and Cross-Validation

To establish a global overview of differences between the EOC patients and the non-cancer controls, multivariate analysis was applied to glycopeptide spectra data as previously described [13,14]. Normalized glycopeptide spectral data sets were unit variance scaled and mean-centered, then analyzed by PCA and OPLS-DA using the SIMCA-P+ program (version 14.1, Umetrics AB; Umeå, Sweden). Model quality was evaluated using R2Y and Q2 values, which reflect the explained fraction of variance and model predictability. PCA was utilized to overview an unsupervised pattern of samples, then OPLS-DA analysis was performed to distinguish two groups, EOC and non-EOC. Before OPLS-DA analysis, the data set was divided into two sets, a training set and a test set, including EOC and non-EOC samples respectively to evaluate the training model’s validity. Separation was evaluated using the first and second principal components taken by PCA or OPLS-DA. OPLS-DA analysis was conducted again for further verification of the differences between the comparison groups.

### 2.7. Study Approval

Institutional Review Board (IRB) approval was obtained at Tokai University (IRB registration number, 09R-082). Signed informed consents were obtained from all participants in the study.

## 3. Results

### 3.1. Selection of Serum Glycopeptides for PCA and OPLS-DA Analysis

We selected 2281 serum glycopeptides for PCA and OPLS-DA analysis from more than 30,000 candidate glycopeptides that were detected as single peaks in the LC–MS analysis of all the serum samples. First, we removed noise peaks by comparing samples with blank-samples. Next, we calculated the inter- and intra-assay coefficents of variability (CVs) of QC samples, which were analyzed every 10 samples, and the glycopeptide peaks with more than 50% of CVs were removed from our final analysis. Then, small peaks whose signal-to-noise ratios were less than 100 also were removed. Finally, isotope, adduct, and fragment ions were deleted from our list of samples for further analysis. As a result of this selection process, the remaining 2281 serum glycopeptides were considered to be sufficiently reliable for the PCA and OPLS-DA analysis.

### 3.2. Comparison of Glycopeptide Data between Early EOC Patients and Leiomyoma and Ovarian Cyst Non-Cancer Patients

Figure 1 and Figure 2 show the comparison of glycopeptide data between early EOC patients and leiomyoma and ovarian cyst non-cancer patients using four different methods of analysis. Figure 1A shows a volcano plot of the serum digested glycopeptide data in early EOC patients compared to leiomyoma and ovarian cyst non-cancer patients. The X-axis corresponds to fold decreases and increases as indicated, and the Y-axis represents the *p*-values from low to high significance. The red and grey dots represent the 2281 glycopeptide peaks that had been selected for analysis. The red dots in the plot are the differentially expressed glycopeptides with statistical significance (*p* < 0.05) and a mean fold >1.5. The X-axis shows mean fold (<1) or inverse of mean fold ratio (>1), while the Y-axis shows the −log_10_ of the adjusted *p* value. The dashed horizontal lines indicate statistical significance of *p* < 0.05.

Figure 1B shows a PCA scatter plot of the 27 leiomyoma and ovarian cyst patient (leiomyoma and ovarian cyst, green dots) and the 23 EOC patient (blue dots) serum samples. We used multivariate analysis to determine consistent variations between data sets. Unsupervised pattern recognition was initially carried out using preliminary PCA to generate an overview of variations between early-stage EOC patients and non-cancer (leiomyoma and ovarian cyst) patients. Clustering based on disease status was not observed on the scores plot of the first two principal components. However, it is evident from this plot that there is a relatively clear separation between the clustering of most of the early stage EOC patients and the non-cancer patients.

Figure 2A shows the OPLS-DA score plot of glycopeptide spectra of serum samples obtained from the 19 early-stage EOC patients and 22 leiomyoma and ovarian cyst non-cancer patients shown in Figure 1B. To assess the predictive ability of the model using unknown samples, about 80% of samples (training set, early-stage EOC *n* = 19, non-cancer patients *n* = 22) were randomly selected to construct an OPLS-DA model, which was then used to predict the remaining 20% (testing set, early-stage EOC *n* = 4, non-cancer patients *n* = 5). Clearly, a better separation of the early-stage EOC patients from the non-cancer (leiomyoma and ovarian cyst) patients was achieved by supervised orthogonal partial least-squares discriminant analysis (OPLS-DA) scores plot. Internal validation was performed to assess the predictive ability of the corresponding OPLS-DA model with the calculated output (*R2*X = (0.21), *R2*Y = (0.84), Q2 (cum) = (0.60)), suggesting the good fit of the model as exceeding the indicator Q2 cutoff of 0.5 (Appendix A).

Figure 2B shows the results of additional external validation set that was not included in Figure 2a with the scores plot of the OPLS-DA prediction model. The testing set of 4 early-stage EOC patients (Figure 2B) were correctly localized to the same region of the training set of early-stage EOC patients (Figure 2A), and equivalent results were obtained using the non-cancer (leiomyoma and ovarian cyst) patient training and testing sets. This confirmed that the serum glycopeptide profiles from the early-stage EOC patients and non-cancer (leiomyoma and ovarian cyst) patients were clearly separated using the OPLS-DA scores plot (Figure 2A,B).

The heat map in Figure 3 shows that the spectra of 300 glycoproteins chosen as an example from the other 2281 peaks on the basis of *p*-values in a *t*-test can differentiate between most patients with EOC (early and advanced) and the leiomyoma and ovarian cyst non-cancer patients. The heat map was constructed by first calculating the averages of the control group (leiomyoma and ovarian cyst) for each glycopeptide. Next, the ratio between each individual’s level and the average was calculated.

Finally, the colors in Figure 3 were assigned for each ratio, whereby the rows show each glycoprotein and the columns show the individual patients. It is evident from this heat map that of the 300 serum glycopeptides we selected for analysis a far greater ratio of glycopeptide peaks exists in the EOC group compared to the non-cancer group. However, some individuals in the cancer group were similar to the non-cancer group and a few individuals in the non-cancer group had high ratio patterns similar to those in the cancer group.

### 3.3. Comparison of Glycopeptide Data between Early Ovarian Clear Cell Carcinoma Patients and Endometrioma Patients

We used the same four methods of data analysis and a heat map generation (Figure 4 and Figure 5) to compare the glycopeptide data between nine early-stage OCCC patients and 18 endometrioma patients as was previously performed for EOC analysis in Figure 2. Thus, Figure 4A is a volcano scatter plot, Figure 4B is a PCA scatter plot, and Figure 5A,B is the score plot of the OPLS-DA model using training and test sets respectively. To assess the predictive ability of the model using unknown samples, about 80% of samples (training set, endometrioma *n* = 14, early-stage OCCC *n* = 7) were randomly selected to construct an OPLS-DA model, which was then used to predict the remaining 20% (testing set, endometrioma *n* = 4, early-stage OCCC *n* = 2). It is evident from a comparison of the Figure 4B and Figure 5A,B that the OPLS-DA scatter and scores plots presented the clearest and best separation between the early-stage OCCC patients and endometrioma patients.

The validation was performed to assess the predictive ability of the corresponding OPLS-DA model with the calculated output (*R2*X = (0.37), *R2*Y = (0.74), Q2 (cum) = (0.44)). Although it did not quite reach the cutoff value for reliable model due to the limited sample size, the result of this subgroup analysis was surely suggestive for the validation of the main study cohort (Q2 > 0.5; Appendix A). The additional external validation set for the endometrioma was correctly localized to the same region as the endometrioma training set, and equivalent results were obtained using the early-stage OCCC testing set (Figure 5B).

The heat map in Figure 6 shows that the spectra of 300 glycopeptides (rows) chosen as an example from the 2281 peaks on the basis of *p*-values in a *t*-test clearly discriminated endometrioma from early-stage OCCC and advanced-staged OCCC in most patients (columns). It is evident from this map that the serum glycopeptides that we selected for this analysis (rows) displayed a far greater ratio of glycoproteins in the two OCCC groups (mostly red bands) than endometrioma group (mostly green bands). However, one individual in the cancer group had a low ratio of glycoproteins similar to the endometrioma group, and a few individuals in the endometrioma group had high ratio patterns similar to those seen in the OCCC group.

## 4. Discussion

In a previous study, we used LC–MS for comparative profiling of serum glycoproteins of ovarian cancer cases and non-cancer controls and found that there were more than 30,000 glycopeptide peaks that could be used potentially as serum biomarkers for the early diagnosis of ovarian cancer [5]. The glycopeptide A2160 was one of the first glycopeptides that we identified as a potential cancer marker and peptide sequencing revealed that it was a component of the fully-sialylated alpha-chain of the complement 4-binding protein. Moreover, we showed that the serum levels of A2160 exhibited superior sensitivity and specificity to CA-125 as an EOC marker. However, many other glycopeptide peaks showed a statistically significant difference between ovarian cancer patients and control patients, suggesting that they also could be used as markers to identify cancer patients.

The aim of the present study was to validate a LC–MS profiling method that we had used previously to determine and visually quantitate multiple serum glycopeptides as a useful tool for the detection and diagnosis of EOC. We investigated the LC–MS quantitative profiling of 2281 glycopeptides selected on the basis of statistical significance following an initial *t*-test of 30,000 glycopeptides detected in the serum of cancer and non-cancer patients. We used the same validated statistical and visualization methods to quantitate the signature glycopeptides in the ovarian cancers compared to two separate non-cancer controls, the endometrioma control group as well as the leiomyoma and ovarian cyst control group. In both cases the signature glycopeptides in the cancer group were highly differentiated from the control groups using various measures including PCA, OPLS-DA scatter and scores plots, and heat maps.

In this study, we did not characterize or identify individual glycopeptides. Instead we used them together as a single group to demonstrate the potential marker signature of ovarian cancer. It is clear from the present study that there are thousands of potential glycopeptide biomarkers for ovarian cancer, but it will take considerable time to isolate, sequence and identify all of them. In the meantime, we have devised an analytical method to measure the individual concentrations of thousands of glycopeptides using a single assay, which substantially reduces analytical time and sample size. In contrast to classical biochemical approaches using immunological methods such as ELISA, RIA and EIA that mainly focus on single targets, our CSGSA involves the collation of quantitative results for a broad series of glycopeptides to reveal an overall change in blood sera of ovarian cancer patients compared to controls. 

As part of our statistical and visual analysis to differentiate between the cancer and non-cancer groups, we used various measures including the volcano plot, PCA, OPLS-DA scatter and scores plots, and heat maps. The volcano plots were used initially to plot significance (*t*-statistic) versus fold-change on the Y and X axes respectively to compare the cancer and non-cancer samples and identify changes in the large glycopeptide data set generated by the LC–MS. From these volcano plots, we identified glycopeptides candidate EOC glycopeptide markers. The unsupervised PCA identified a set of unique spectral patterns that generally separated the cancer and non-cancer data into relatively distinct clusters. However, some of the cancer spectral data overlapped with the non-cancer data.

In comparison to PCA, the supervised OPLS-DA scatter plots clearly forced a greater scores-space separation that grouped the cancer and non-cancer data into two significantly distinct clusters. Worley and Powers [13] used Monte Carlo simulations to demonstrate that the PCA and the OPLS-DA analyses should be performed together on the same samples to confirm reliability because the OPLS-DA analysis alone can yield statistically unreliable group separation. They found that when the PCA failed to expose group separation, OPLS-DA continued to separate the groups at the expense of model reliability. Therefore, we used both the PCA and the OPLS-DA analyses to test and confirm the reliability of our results. We further validated the reliability of our results by computing OPLS-DA scatter and scores plots in association with the OPLS-DA prediction model.

Heat maps are another efficient way to visualize complex data sets organized as matrices. Our study is the first to report on the MS-based glycopeptides quantitative levels in ovarian cancer and non-cancer individual patients using statistical comparative analysis with heat maps. We used heat maps to visualize the quantitative glycopeptide data in individual patients as a single graphic that allowed us to easily identify the signature patterns of many glycopeptides and their quantities across multiple subjects in highly complex MS datasets comparing cancer and non-cancer patients. The relative concentrations of the glycopeptides were statistically filtered between the patients and controls and the quantitative results were displayed for each individual to reveal the most effective glycopeptide signatures. Although most of the statistically significant glycopeptides showed higher levels in the patients with cancer, some individuals had levels similar to those in the non-cancer group. These inconsistencies may have arisen from the fact that individual ratios were based on the average glycopeptide levels in the non-cancer group and not on the potentially variable glycopeptide base-line levels of each individual. However, the heat maps seem to reflect the trends of the PCA results more closely than the OPLS-DA. These sample outliers would need to be followed up in greater detail to better understand the inconsistencies between the cancer and non-cancer groups and between the PCA and OPLS-DA results.

We have used the mass spectra profiles of serum glycopeptides as multi-biomarkers for the purpose of cancer screening. Kim et al. [17] previously had undertaken a serum multi-biomarker MS screening of ovarian cancers, but they screened for N-glycans instead of glycopeptides. No study has previously used the CSGSA for the detection of ovarian cancer. However, like Kim et al [17], we focused on diagnosis rather than marker discovery and even though at this stage we do not understand the biological context of the changes, the MS signals exhibited differences between the cancer and non-cancer groups that could be exploited to improve current cancer screening procedures.

Our study shows that CSGSA is a potentially new tool for the early detection of ovarian cancer using glycopeptides within blood serum samples that could also be applied for the detection and diagnosis of other cancers. We believe that the use of multiple biomarkers can lead to far better diagnostic classification than current single-marker approaches [11,17] that depend on CA-125 or HE4. Our analysis is early and many more comprehensive studies are required to determine if the over-expressed blood serum glycopeptide markers are specific to certain cancers such as EOC or if they also markers for various other cancer and pre-cancer types. Thus, we will need to ascertain the specificity of our markers to ovarian cancer by comparing our current results in patients with ovarian cancer to those with other cancer types. We must also further address how these markers change with different stages of precancerous and cancerous lesions.

Another limitation in our study is small sample size. Generally, high-grade serous ovarian cancer is the most common histology type of ovarian cancer, and therefore, ovarian cancer screening to distinguish stage I high-grade serous ovarian cancer from benign ovarian tumors is the most clinically meaningful comparison. However, this comparison was not feasible as there were too few cases of this subgroup to analyze in our dataset. Moreover, comparisons between histology types were also not feasible. In Japan, OCCC is a particularly common histology, presenting typically at an early-stage [18]. As women with endometrioma possess an increased risk of OCCC [19], comparison of early-stage OCCC to endometrioma is more relevant to the Japanese population.

## 5. Conclusions

CSGSA described in this study may provide important signatures and biomarkers for ovarian cancer patients. Early detection and subtype determination prior to surgery are crucial for clinicians to design an effective treatment strategy for each patient, fitting with the goal of precision medicine.

## Figures and Tables

**Figure 1 cancers-11-00591-f001:**
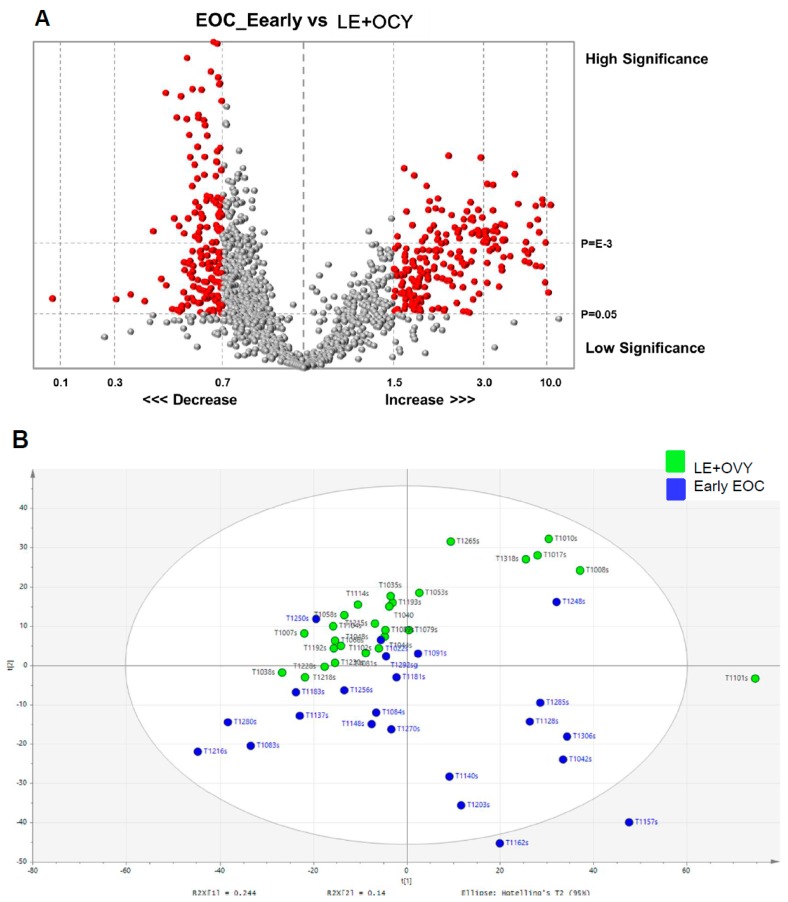
Glycopeptide spectral data in an EOC group and a non-cancer group. (**a**) Volcano plot of serum digested glycopeptide data in early EOC patients compared to non-cancer (LE and OVY) patients. The Y-axis is the *p*-values determined by the t-test of the mean fold change and the X-axis is the mean fold change. The red dots in the plot represents the differentially expressed glycopeptides with statistical significance (*p* < 0.05) and mean fold change of <0.7 or >1.5. The vertical lines in the matrix represent the log_2_ fold changes and the horizontal lines in the matrix represent the statistical significance of *p* < 0.05. (**b**) PCA scatter plot of LE and OVY (green dots) and EOC patient (blue dots) serum samples. The ellipse represents 95% confidence interval. Abbreviations: EOC, epithelial ovarian cancer; LE, leiomyoma; and OVY, ovarian cyst.

**Figure 2 cancers-11-00591-f002:**
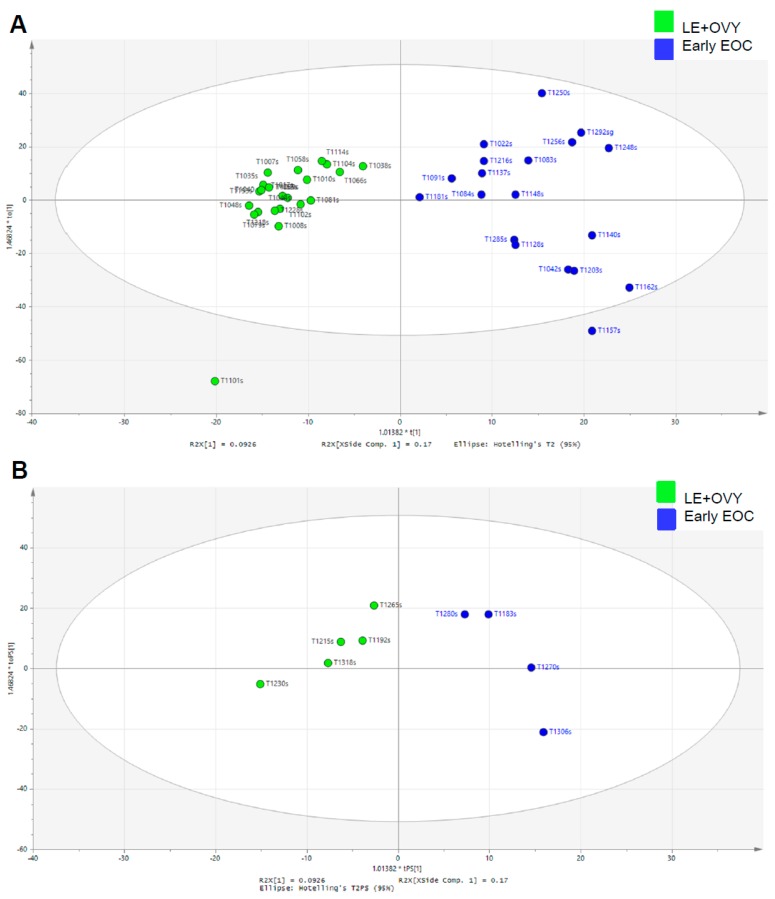
Glycopeptide spectral data in a cancer group and a non-cancer group. (**A**) OPLS-DA scatter plot based on the same samples as in panel **B**. Green dots are LE and OVY samples and blue dots are EOC patients. The ellipse represents 95%CI. (**B**) Scores plots of the OPLS-DA prediction model for validation set. Green dots are LE and OVY samples and blue dots are EOC patients. The ellipse represents 95% confidence interval. Abbreviations: EOC, epithelial ovarian cancer; LE, leiomyoma; and OVY, ovarian cyst.

**Figure 3 cancers-11-00591-f003:**
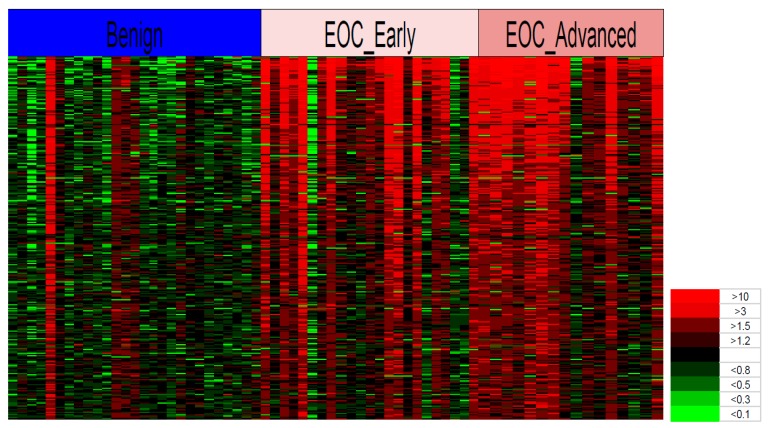
Heat map of glycopeptides in the blood sera of early-stage EOC, advanced-stage EOC, and non-cancer patients. The Y-axis represents 300 individual glycopeptides in the MS spectra and the X-axis represents the individual patients in the cancer and non-cancer groups. The spectral colors in the matrix represent the ratio level of glycopeptide peaks in the individual compared to the average level in the non-cancer group ranging from the brightest green (<0.1) to the brightest red (>10). Abbreviation: EOC, epithelial ovarian cancer.

**Figure 4 cancers-11-00591-f004:**
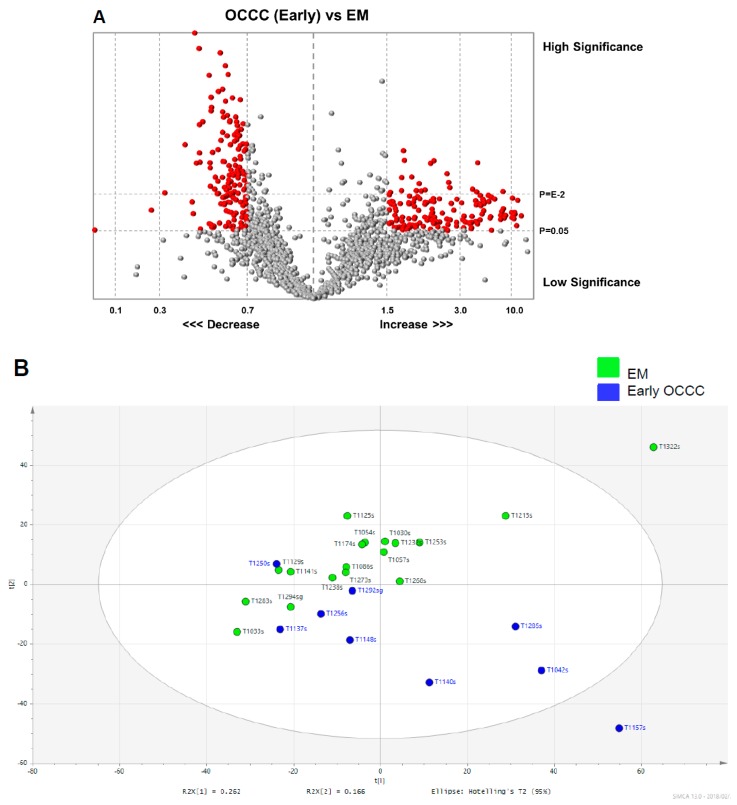
Glycopeptide spectral data in early-stage OCCC group and endometrioma group. (**A**) Volcano plot of serum digested glycopeptide data in early-stage OCCC *n* = 9 compared to 18 endometrioma patients. The Y-axis is the *p*-values determined by the *t*-test of the mean fold change and the X-axis is the mean fold change. The red dots in the plot represents the differentially expressed glycopeptides with statistical significance (*p* < 0.05) and mean fold change of <0.7 or >1.5. The vertical lines in the matrix represent the log_2_ fold changes and the horizontal lines in the matrix represent the statistical significance of *p* < 0.05. (**B**) PCA scatter plot of endometrioma (green dots) and early-stage OCCC patient (blue squares) serum samples. The ellipse represents 95% confidence interval. Abbreviations: OCCC, ovarian clear cell carcinoma; and EM, endometrioma.

**Figure 5 cancers-11-00591-f005:**
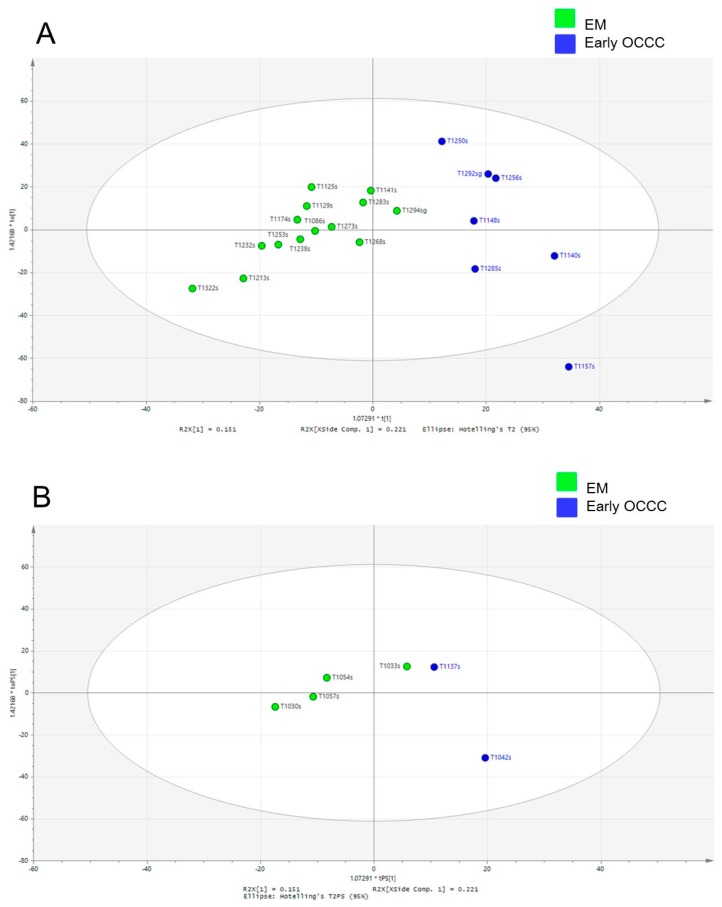
Glycopeptide spectral data in early-stage OCCC group and endometrioma group. (**A**) OPLS-DA scatter plot based on the same samples as in panel B. Green dots are endometrioma samples and blue dots are early-stage OCCC patients. The ellipse represents 95%CI. (**B**) Scores plots of the OPLS-DA prediction model for validation set. Green dots are endometrioma samples and blue dots are early-stage OCCC. The ellipse represents 95% confidence interval. Abbreviations: OCCC, ovarian clear cell carcinoma.

**Figure 6 cancers-11-00591-f006:**
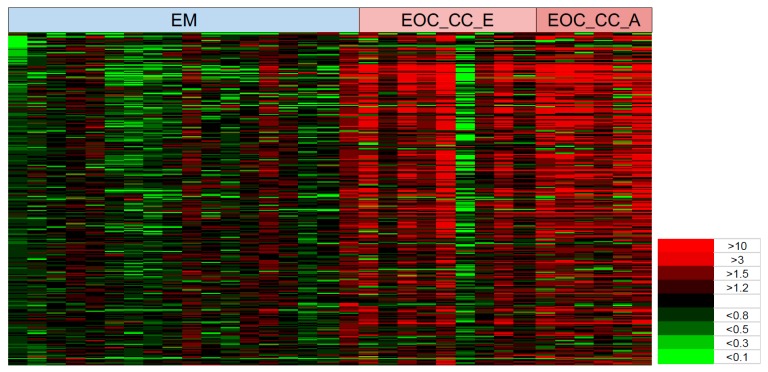
Heat map of glycopeptides in the blood sera of early OCCC, advanced OCCC and endometrioma patients. The *Y*-axis represents 300 individual glycopeptides in the MS spectra and the *X*-axis represents the individual patients in endometrioma, early OCCC, and advanced-stage OCCC patient groups. The spectral colors in the matrix represent the ratio level of glycopeptide peaks in the individual compared to the average level in the non-cancer group ranging from the brightest green (<0.1) to the brightest red (>10). Abbreviation: OCCC, ovarian clear cell carcinoma; EM, endometrioma; EOC_CC_E, early-stage OCCC; and EOC_CC_A, advanced-stage OCCC.

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
