# Peer review of "Comprehensive Serum Glycopeptide Spectra Analysis (CSGSA): A Potential New Tool for Early Detection of Ovarian Cancer"

_cancers, 2019, doi:10.3390/cancers11050591_

Round 1

Reviewer 1 Report

This study analyzed differential glycopeptide peak spectra between cancer and non -cancer groups by PCA, OPLS-DA, and heat maps.

This is very interesting subjects and it is worthwhile subject because there are not many research results in this field.

However, the researches only analyzed the expression pattern, there is no quantitative data.

It is need to add quantitative results of some glycopeptides or validation set's results.

Author Response

Reply to Reviewer 1.

This study analyzed differential glycopeptide peak spectra between cancer and non -cancer groups by PCA, OPLS-DA, and heat maps.

This is very interesting subjects and it is worthwhile subject because there are not many research results in this field.

However, the researches only analyzed the expression pattern, there is no quantitative data.

R1-1. It is need to add quantitative results of some glycopeptides or validation set's results.

Reply: Line 212-218 and 252-257, Figure 2B and 5B, and Supplemental Figure S1.

We have tested multiple validation sets with results shown in the above referred sections. Per the suggestion, we have clarified the contents in the revised manuscript. We have also provided the results of validation set in supplemental file.

Reviewer 2 Report

This is a well written paper and as noted early diagnosis is key for ovarian ca detection AND for ultimate prognosis.  Ideally, I would have liked to have seen more serous carcinoma given this is the most common type of ovarian cancer subtype.  The significant inclusion of clear cell and endometriod subtypes might actually skew results for the authors.

Therefore, I would LIKE to see an additional analysis with the serous carcinoma alone compared to benign in a similar manner to the OCCC analysis (and perhaps even adding additional cases and then reanalyzing if feasible) to determine the potential bias that is incorporated (or NOT!).  Additionally, an analysis comparing AMONG Ovarian CA subtypes would be beneficial.

I would also like to see a better demographic table further characterizing the stage based on subtype.

Author Response

Reply to Reviewer 2.

This is a well written paper and as noted early diagnosis is key for ovarian ca detection AND for ultimate prognosis.  Ideally, I would have liked to have seen more serous carcinoma given this is the most common type of ovarian cancer subtype.  The significant inclusion of clear cell and endometriod subtypes might actually skew results for the authors.

R2-1. Therefore, I would LIKE to see an additional analysis with the serous carcinoma alone compared to benign in a similar manner to the OCCC analysis (and perhaps even adding additional cases and then reanalyzing if feasible) to determine the potential bias that is incorporated (or NOT!).  Additionally, an analysis comparing AMONG Ovarian CA subtypes would be beneficial.

Reply: Line 423-430

We agree with the reviewer that comparison of stage I HGSOC to benign ovarian tumors would be the most ideal comparison for ovarian cancer screening. In our study, unfortunately, this comparison was not feasible as there were few cases to analyze. Moreover, due to the small sample size, comparisons between histology types were also not feasible. In Japan, OCCC is particularly common, presenting generally at an early-stage. As Japanese women with endometrioma carry a significantly increased risk of OCCC, comparison of early-stage OCCC to endometrioma is more relevant in our population. We have discussed this limitation in the revised manuscript. We are currently conducting a secondary study with a larger sample size, where we plan to address these limitations.

R2-2. I would also like to see a better demographic table further characterizing the stage based on subtype.

Reply: Supplemental Table S1

We have provided more detail for the demographic table as suggested by the reviewer.

Round 2

Reviewer 1 Report

The revised manuscript reflected well reviewer's comments.

Minor English language spell check required.

Reviewer 2 Report

As noted, I would ideally like additional analysis and I understand that this is not available in the present manuscript.  Given further studies are underway I feel comfortable allowing this to move forward.